# Screening for Virulence-Related Genes via a Transposon Mutant Library of *Streptococcus suis* Serotype 2 Using a *Galleria mellonella* Larvae Infection Model

**DOI:** 10.3390/microorganisms10050868

**Published:** 2022-04-21

**Authors:** Jingyan Fan, Lelin Zhao, Qiao Hu, Siqi Li, Haotian Li, Qianqian Zhang, Geng Zou, Liangsheng Zhang, Lu Li, Qi Huang, Rui Zhou

**Affiliations:** 1State Key Laboratory of Agricultural Microbiology, College of Veterinary Medicine, Huazhong Agricultural University, Wuhan 430070, China; fjy6168@webmail.hzau.edu.cn (J.F.); zhaolelin@webmail.hzau.edu.cn (L.Z.); huqiao@webmail.hzau.edu.cn (Q.H.); siqil5@illinois.edu (S.L.); lht@webmail.hzau.edu.cn (H.L.); qianzhang2022@163.com (Q.Z.); zougeng918@webmail.hzau.edu.cn (G.Z.); zls2015@webmail.hzau.edu.cn (L.Z.); lilu@mail.hzau.edu.cn (L.L.); 2International Research Center for Animal Disease (Ministry of Science & Technology of China), Wuhan 430070, China; 3Cooperative Innovation Center of Sustainable Pig Production, Wuhan 430070, China; 4The HZAU-HVSEN Research Institute, Wuhan 430042, China

**Keywords:** *Streptococcus suis*, transposon mutant library, virulence-related genes, *Galleria mellonella* larvae, *hxtR*

## Abstract

*Streptococcus suis* (*S. suis*) is a zoonotic bacterial pathogen causing lethal infections in pigs and humans. Identification of virulence-related genes (VRGs) is of great importance in understanding the pathobiology of a bacterial pathogen. To identify novel VRGs, a transposon (Tn) mutant library of *S. suis* strain SC19 was constructed in this study. The insertion sites of approximately 1700 mutants were identified by Tn-seq, which involved 417 different genes. A total of 32 attenuated strains were identified from the library by using a *Galleria mellonella* larvae infection model, and 30 novel VRGs were discovered, including transcription regulators, transporters, hypothetical proteins, etc. An isogenic deletion mutant of *hxtR* gene (Δ*hxtR*) and its complementary strain (CΔ*hxtR*) were constructed, and their virulence was compared with the wild-type strain in *G. mellonella* larvae and mice, which showed that disruption of *hxtR* significantly attenuated the virulence. Moreover, the Δ*hxtR* strain displayed a reduced survival ability in whole blood, increased sensitivity to phagocytosis, increased chain length, and growth defect. Taken together, this study performed a high throughput screening for VRGs of *S. suis* using a *G. mellonella* larvae model and further characterized a novel critical virulence factor.

## 1. Introduction

*Streptococcus suis* (*S. suis*) is an important zoonotic pathogen causing serious diseases in pigs, including meningitis, endocarditis, and sepsis, and can also infect humans, leading to streptococcal toxic shock-like syndrome (STSLS) and even acute death [1,2,3]. Serotype 2 (SS2) is the most virulent and prevalent serotype in pigs and human beings among the numerous (at least 29) serotypes of *S. suis* [4]. More than one thousand cases of human *S. suis* infections have been reported all over the world, and two large-scale outbreaks in China led to 55 deaths of people and huge economic losses to the pig industry [5].

For decades, continuous efforts have been concentrated on studying the virulence of SS2, and numerous virulence factors (VFs) have been discovered, including suilysin (Sly) [6,7], capsular polysaccharide (CPS) [8], and arginine deiminase system (ADS) [9], etc. Nevertheless, the pathogenesis of *S. suis* remains to be unraveled, which reveals the necessity of the continued search for novel VFs and virulence-related genes (VRGs) in *S. suis*.

Transposon (Tn) is a mobile DNA element that can change its position in a genome. Insertion at the coding sequence or the promoter region of a gene can cause disruption of the gene. By using this feature, a large-scale random mutagenesis library can be generated efficiently. Tn mutagenesis has been extensively applied to identify functional genes in many species, including *S. suis* [10], *Mycoplasma bovis* [11], *Pseudomonas aeruginosa* [12], *Salmonella* Gallinarum [13], and *Aeromonas hydrophila* [14], etc.

Infection models are critical for identifying bacterial virulence factors. Various assays have been developed for virulence traits evaluation, including whole blood bactericidal assay, phagocytosis assay, adhesion and invasion assay, and classical animal infection models. However, the in vitro and ex vivo systems can not sufficiently reflect the complex environment in the host. Animal infection models, including pig [15], mouse [16], and zebrafish [17,18], provide valuable platforms to study the mechanism of pathogenesis of *S. suis*. However, they are time-consuming, expensive, and not animal-friendly. Therefore, a convenient and reliable surrogate animal model to rapidly assess the virulence of numerous *S. suis* mutants is urgently needed. Wax moth (*Galleria mellonella*) larvae have successfully served as a model to estimate the virulence of *Streptococcus* spp. [19,20], including *S. suis* [21]. It complies with the 3R principle of animal experiments.

Here, we constructed a Tn mutant library of *S. suis* and sought to identify more novel VRGs by screening a Tn library using the *G. mellonella* larvae infection model. A total of 32 mutants were identified that showed significantly attenuated virulence, involving 32 genes. A previously unreported gene encoding an XRE family transcriptional regulator HxtR was further selected and characterized.

## 2. Materials and Methods

### 2.1. Strains and Plasmids

The bacterial strains, plasmids, and primers used in this study are listed in Table 1. *S*. *suis* SC19 strain is a virulent strain of serotype 2 isolated from the 2005 *S*. *suis* outbreak in Sichuan Province, China [22]. Plasmid pTV408 containing a Tn917 transposon was kindly donated by Dr. Tracy Wang (University of Cambridge, Cambridge, UK); it was used for Tn mutagenesis in *S. suis*. *S. suis* SC19 and its derivatives were grown in tryptic soy broth (TSB; BD, Franklin Lakes, NJ, USA) or on tryptic soy agar (TSA; BD, Franklin Lakes, NJ, USA) with 10% (*v*/*v*) fetal bovine serum (FBS; Sijiqing, Hangzhou, China) at 37 °C. Erythromycin (1 µg/mL) was added to screen the transposants, and spectinomycin (100 µg/mL) was used to select a complemented strain. *E. coli* DH5α serving as the host strain for cloning was cultured in lysogeny broth (LB) or plated onto LB agar at 37 °C.

### 2.2. Construction of Tn Library

The transformant containing plasmid pTV408 was obtained by electrotransformation of pTV408 into the prepared competent cells of *S. suis* SC19, as described previously [25]. Loss of plasmid and integration of transposon was achieved through subsequent incubation of the transformants on CBA plates containing 5% (*v*/*v*) defibrinated sheep blood (DfSB; Yiqi, Zhengzhou, China) and 1 μg/mL erythromycin at the non-permissive temperature (37 °C) for the plasmid [26]. After five times incubation onto fresh CBA plates containing erythromycin at 37 °C, the erythromycin-resistant and kanamycin-sensitive clones were obtained, and transposon mutants were determined by PCR with primer pair JPM48/JPM49 to detect the presence of erythromycin-resistance cassette within the transposon and primer pair JPM50/JPM51 to detect the presence of kanamycin-resistance cassette located at the backbone of pTV408 plasmid (Table 1). All the erythromycin resistance cassette-positive and kanamycin resistance cassette-negative colonies (Tn library) were grown in TSB for 6–8 h and stored at −80 °C in 25% (*v*/*v*) glycerol.

### 2.3. Identification of Insertion Sites in Tn Library

The genomic DNA was extracted from each Tn mutant using the E.Z.N.A.^®^ Bacterial DNA kit (Omega, Norcross, GA, USA), and the Tn insertion site was determined by using a linker PCR. Briefly, 25 µg genomic DNA was digested with 3 units of *Alu*I for 1 h at 37 °C and cleaned up with a MiniElute PCR Purification kit (Qiagen, Hilden, Germany). The linker primers 254/256 were annealed via incubation at 95 °C for 2 min and slowly cooled on bench, and then, they were ligated to the digested genomic DNA using a DNA Ligation Kit Ver.2.1 (Takara, Japan) at 16 °C for at least 2 h. The ligation product was cleaned up with QiaQuick PCR purification kit (Qiagen, Hilden, Germany). Eventually, linker PCR with another pair of primers Tn917-seq/258 was performed, and all products were subjected to DNA sequencing by Quintara Bio, Wuhan, China. The Tn insertion sites were determined by mapping the sequencing results of the PCR products to the genome sequence of *S. suis* SC19 (GenBank accession number NZ_CP020863).

CGView Server was used to visualize the distribution of the inserted genes involved in the mutant library on the *S. suis* SC19 genome [27]. For KEGG pathway enrichment, BlastKOALA tool was first used to assign the K number to each gene [28]. The K number was then used to map each gene to the corresponding KEGG pathway using the Reconstruct tool of the KEGG mapper [29].

### 2.4. Screening the Virulence Attenuated S. suis Mutants Using G. mellonella Larvae

Mutants were revived by growing two generations on a TSA plate with 10% FBS and 1 μg/mL erythromycin at 37 °C and diluted to 0.5 MacFarland (1.5 × 10^8^ CFU/mL) using physiologic saline. The bacterial concentration of mutants was measured by using BD PhoenixSpec Nephelometer (BD, Franklin Lakes, NJ, USA). *G. mellonella* larvae, purchased from Yi Jia Yi Insect Breeding Ltd., were stored in the dark at 15 °C. The larvae with a weight between 0.4 and 0.5 g were injected with 20 μL of bacterial suspensions (approximately 3 × 10^6^ CFU) into the left posterior proleg, and the same amount of *S. suis* SC19 cells and physiologic saline were used as the positive and negative control, respectively. Six larvae were used for each group for the first-round screening. The survival of the larva was monitored at 12 h, 18 h, and 24 h post-infection (hpi).

### 2.5. Construction of hxtR Deletion Mutant and Complemented Strain

To construct *hxtR* deletion mutant, homologous recombination was performed as previously reported [23]. Briefly, the upstream and downstream fragments flanking of gene *hxtR* coding sequence (B9H01_RS05155) were amplified from the *S. suis* SC19 genomic DNA with primer pairs Δ*hxtR*-A/Δ*hxtR*-B and Δ*hxtR*-C/Δ*hxtR*-D, respectively (Table 1), and were cloned into plasmid pSET4s shuttle vector [23] through seamless cloning. Recombinant plasmids were electroporated into the *S. suis* SC19 strain, and the isogenic deletion strains were selected by double homologous recombination as described previously [30].

For complemented strain construction, the coding sequence and promoter of gene *hxtR* were amplified using primer pair CΔ*hxtR*-F/CΔ*hxtR*-R (Table 1) and cloned into the pSET2 vector [24]. The recombinant plasmid pSET2:*hxtR* was subsequently transformed into Δ*hxtR* to generate the complemented strain CΔ*hxtR*.

To further confirm the mutant strain Δ*hxtR* and the complemented strain CΔ*hxtR*, RNA was extracted from each strain using SV Total RNA Isolation System (Promega, Madison, WI, USA) according to the manufacturer’s instructions, and cDNA was synthesized using PrimeScript^TM^ RT reagent Kit (Takara, Kusatsu, Japan). The cDNA was used as the template to determine the expression of *hxtR* and its upstream and downstream genes by reverse transcription (RT)-PCR analysis with primer pairs *hxtR*-F/*hxtR*-R, 5150-F/5150-R, and 5160-F/5160-R, respectively (Table 1).

### 2.6. Mouse Infection Experiment

To further confirm the virulence results of Δ*hxtR* in *G. mellonella* larvae, a mouse infection model was used. Six-week-old specific-pathogen-free (SPF) Kunming mice with similar body weights (18~22 g) were randomly grouped into three groups (n = 8) and intraperitoneally infected with 8.5 × 10^8^ CFU of *S. suis* SC19 and Δ*hxtR*. A negative control group was set, which was injected with the same volume of physiological saline. Mouse survival after infection was recorded every 24 h for 7 d.

### 2.7. Whole Blood Bactericidal Assay

The whole blood bactericidal assay was modified from the previous report [31]. Cultures of *S. suis* SC19, Δ*hxtR*, and CΔ*hxtR* in the mid-log phase were washed two times using physiological saline, then diluted to about 1~5 × 10^7^ CFU/mL. A total of 50 μL of bacterial suspensions was mixed with 450 μL fresh mouse blood and incubated at 37 °C for 30 min. The mixtures were serially diluted and plated on TSA plates. Bacterial survival was calculated as (CFU recovered/CFU in original inoculum) × 100%.

### 2.8. Phagocytosis Assay

The phagocytosis assay was performed according to a previous study [32]. Briefly, RAW264.7 murine macrophage cells were cultured at 37 °C and 5% CO_2_ overnight in RPMI 1640 medium with 10% FBS (Gibco, Invitrogen, Carlsbad, CA, USA) to form monolayers in 12-well plates with each well containing about 1.5 × 10^6^ cells. Bacterial cells of *S. suis* SC19, Δ*hxtR*, and CΔ*hxtR* at the mid-log phase growth were added to the wells at a multiplicity of infection (MOI) of 10:1. After incubation at 37 °C for 30 min, the infected cells were washed, and fresh medium containing ampicillin (50 μg/mL) was added, followed by incubation for 1 h at 37 °C. Then, the cells were washed three times with phosphate-buffered saline (PBS) after the ampicillin treatment, and phagocytized bacterial cells were determined by lysing the RAW264.7 cells with sterile water and counting the number of released bacteria on TSA plates.

### 2.9. Morphology Observation

The indicated *S. suis* strains of overnight-grown culture were subcultured into fresh TSB and grown to the mid-log growth at 37 °C. For Gram staining, the cells were collected and washed three times with PBS and mounted on glass slides by flaming. The bacterial cells were then imaged following regular Gram staining under a light microscope.

For further morphological characterization, the bacterial cells were washed with PBS and resuspended in PBS containing fluorescent dye Alexa Fluor 647 (AF-647; Thermo, Waltham, MA, USA) at a final concentration of 20 nM and incubated at 37 °C for 30 min. Then, the cells were washed and imaged using a two-color structure illumination microscope (SIM; Nikon Instruments, Tokyo, Japan) with excitation at 651 nm and emission at 672 nm.

For chain length analysis, more than one hundred chains per sample were randomly selected in different fields to measure the length of the chain by using the software Image-Pro, and then, the average chain length was calculated (in µm) [33].

### 2.10. Growth Measurements

To evaluate the impact of *hxtR* knockout on the growth of *S. suis*, a Bioscreen C optical growth analyzer (Lab Systems Helsinki, Vantaa, Finland) was utilized to monitor the growth rates of *S. suis* SC19, Δ*hxtR*, and CΔ*hxtR*. The overnight cultures of the strains were diluted to an initial OD_600_ of 0.01 with TSB, and then, the diluted culture was pipetted into a microplate. The plate was incubated at 37 °C with continuous shaking for 14 h, and the optical density at 600 nm was monitored every 1 h.

### 2.11. Statistical Analysis

Statistical differences between two groups in the whole blood bactericidal assay, phagocytosis assay, and chain length were analyzed using the Student’s *t*-test (unpaired, two-tailed) with GraphPad prism 7. The statistical difference between two groups in the *G. mellonella* larvae and mice infection assays were determined using the log-rank (Mantel–Cox) test with GraphPad prism 7.

## 3. Results

### 3.1. Mutants Construction and Insertion Sites Mapping

By using the Tn917 transposition system, a total of 1665 transposon mutants of *S. suis* were obtained and sequenced individually. The insertion sites were mapped to the genome of *S. suis* SC19 (GenBank accession number NZ_CP020863). As shown in Figure 1A, most insertion sites were located within the region ranging from 800 kbp to 1200 kbp of the genome. Among them, 1191 mutants contained a Tn insertion within the coding sequence of a gene, leading to the disruption of 417 different genes (Appendix A), and 474 mutants contained an insertion in the intergenic region. The genes with a Tn insertion were shown on the genome (Figure 1B). These genes were distributed to 22 different KEGG pathways (Figure 2). 

### 3.2. Virulence Assessment in G. mellonella Larvae

To do high throughput virulence evaluation with the Tn mutants, a *G. mellonella* larva infection model was used. Apart from the Tn mutants that were unable to be revived from the library, a total of 391 mutants were subject to the *G. mellonella* larva infection assay. The results showed that 32 mutants exhibited significantly attenuated virulence compared with the wild-type strain (Table 2). Among these genes, *arcC* and *sspA* were already reported virulence genes. These 32 strains were grouped into four categories according to their genetic loci, including 2 transcription regulators, 3 transporters, 6 hypothetical proteins, and 21 others (Table 2). Among these 32 genes, a novel transcription regulator gene *hxtR* (B9H01_RS05155) was selected for further study.

### 3.3. Decreased Virulence of ΔhxtR in Larvae and Mice

A deletion mutant of *hxtR* (Δ*hxtR*) and its complemented strain CΔ*hxtR* were constructed and verified by PCR and RT-PCR (Appendix A). The RT-PCR results indicated that the knockout of *hxtR* did not affect the *hxtR* downstream gene transcription but led to the increased transcription of the upstream gene. Compared with *S. suis* SC19 and CΔ*hxtR*, Δ*hxtR* showed a significant decrease in virulence in the *G. mellonella* larvae infection assay (Figure 3A).

To further confirm the virulence attenuation of Δ*hxtR*, mouse infection assays were performed. It is shown in Figure 3B that all the *S. suis* SC19-infected mice exhibited obvious clinical symptoms, and most mice (six out of eight) died on the first day, and one died within 2 days post infection. However, only one mouse infected with Δ*hxtR* died within 5 days post infection, and the other mice (seven out of eight) survived until the end of the experiment (Figure 3B), indicating a dramatic decrease in virulence due to the deletion of *hxtR*.

### 3.4. Reduced Resistant Abilities of ΔhxtR to Whole Blood Killing and Phagocytosis

The whole blood killing and phagocytosis assays were performed with Δ*hxtR*, CΔ*hxtR*, and SC19 strains. As shown in Figure 3C, the SC19 strain grew by 206% in mouse blood, while Δ*hxtR* only grew by 110%, indicating that deletion of *hxtR* resulted in a decreased ability to survive in blood. In the phagocytosis assay, compared to the CΔ*hxtR* and SC19 strains, the Δ*hxtR* strain was much easier to be phagocytized by the macrophage cells, indicating an important role of *hxtR* in resisting phagocytosis by RAW264.7 macrophages (Figure 3D).

### 3.5. hxtR Is Involved in Cell Morphology and Growth of S. suis

The impact of *hxtR* deletion on the morphology of *S. suis* was explored by Gram staining (Figure 4A) and Alexa Fluor 647 staining (Figure 4B). The results showed that *hxtR* inactivation induced extensive chain elongation and nearly spherical cells (Figure 4A–C). Subsequently, the growth of the strains was tested using a Bioscreen C optical growth analyzer. Growth curves showed that Δ*hxtR* grew more slowly during the exponential growth phase than SC19 and CΔ*hxtR* in TSB. The Δ*hxtR* reached only half of the optical density value at the stationary phase of the SC19 and CΔ*hxtR* (Figure 4D).

## 4. Discussion

As an emerging zoonotic bacterial pathogen, *S. suis* can not only result in great economic losses in the pig industry [1] but also be transmitted to human beings through direct contact with high mortality [2]. VRGs assist bacterium against the defense of the mucosal immune, adhere to and invade the mucosal epithelial cells, survive in the bloodstream, and eventually attack multiple organs resulting in severe systemic diseases [38]. Therefore, elucidating the molecular mechanisms for *S. suis* pathogenesis is crucial for the development of novel and effective prophylactic and therapeutic approaches.

In order to identify novel VRGs of *S. suis*, a Tn mutant library of *S. suis* was generated. First, the Tn-inserted genes of 1665 mutants in the library were one-by-one sequenced, and the results revealed that most insertion sites were in the 800–1200 kbp region of the *S. suis* SC19 genome. The insertion of Tn917 was not completely random due to the choice of target determined by structural features of DNA, which has been observed in the *S. equi* Tn917 transposant pool [26,39]. Although there is a preference for insertion sites, the 417 different inserted genes identified from the 1665 mutants were scattered throughout the genome and involved in 22 KEGG pathways. Based on these analyses, we concluded that the mutant pool deserved further functional genomics research.

Efficient and appropriate experimental settings are important for gene function studies, especially when high throughput screening is needed. In recent decades, the use of invertebrates to explore microbial pathogenesis has made important contributions to biomedical research. In this study, we screened 391 mutants involved in different genes using the *G. mellonella* larvae infection model, and 32 mutants (32 gene loci) exhibited significantly attenuated virulence compared with *S. suis* SC19. Among these genes, two genes have been discovered to affect the virulence of *S. suis*. *arcC,* encoding a carbamate kinase ArcC, is considered a virulence-associated gene [37]. ArcA, ArcB, and ArcC compose the arginine deiminase system (ADS), which participates in the arginine metabolism [9]. In *S. suis*, the ADS has been reported previously to be involved in the pathogenesis and adaption to acidic conditions [9]. The *sspA* gene encodes a cell wall-associated subtilisin-like serine protease that is also critical for the virulence of *S. suis*. The deletion of *ssp*A exhibited attenuated virulence compared with wild strain in pigs [35,40].

In addition to *arcC* and *sspA,* of which their roles in *S. suis* pathogenesis have been reported, the remaining 30 genes were identified for the first time to be related to the virulence of *S. suis*. Some of these genes, however, have been shown to be involved in the pathogenesis of other bacterial pathogens. For instance, *hylA* encodes a hyaluronidase, and its deletion mutant has been demonstrated to be significantly attenuated using a murine model in *Staphylococcus aureus* [41]. The formate C-acetyltransferase (PFL) is involved in the pyruvate metabolism, which has been shown to lead to a change in the lipid composition of the cell membrane and attenuate the virulence in mice when disrupted in *Streptococcus pneumoniae* [42]. These data suggest that our Tn library screening provides some novel candidate VRGs deserving further study in *S. suis*.

*hxtR* encodes an XRE family transcriptional regulator, and its function is unknown in *S. suis*. It was identified to influence the virulence of *S. suis* in our primary Tn library screening. To further confirm this, a deletion mutant of *hxtR* and its complemented strain were constructed. Independent *G. mellonella* larvae and mice infection assay validated that *hxtR* played an important role in the pathogenesis of *S. suis*. Our subsequent in vitro studies revealed that Δ*hxtR* displayed a reduced growth ability in whole blood and increased sensitivity to phagocytosis, further confirming the virulence phenotype. However, no significant change in the abilities to adhere to and invade porcine kidney-15 cells was observed when compared with the wild-type and complementary strains (data not shown). Moreover, the morphological study suggested that Δ*hxtR* presented abnormal cell division manifested by the formation of exceptional long chains of bacterial cells. Considering that bacterial cell division is one of the most important physiological processes, a possible explanation for the attenuated virulence of Δ*hxtR* could be attributed to its influence on the replication and growth of *S. suis*.

XRE family transcriptional regulators are widespread in bacteria, and they are involved in the regulation of virulence, antibiotic synthesis and resistance, and stress response [43,44]. In *S. suis*, there are multiple XRE family transcriptional regulators, among which SrtR has been previously confirmed to regulate the virulence in a murine model and oxidative stress tolerance [45], and XtrSs was shown to be involved in response to hydrogen peroxide stress [46]. Moreover, XRE family transcriptional regulators were shown to participate in the direct regulation of virulence factors. In *Staphylococcus aureus*, the XRE family transcriptional regulator XdrA was reported to control the expression of the virulence gene *spa* [43]. In group B *streptococcus*, XtgS was demonstrated to negatively regulate its virulence by directly repressing *pseP* transcription [47]. In our study, deletion of *hxtR* seemed to interfere with the expression of its upstream gene, which is a hypothetical gene. Whether this influence contributes to attenuated virulence of Δ*hxtR* needs further investigation. Furthermore, identifying the regulation targets will be crucial to revealing the underlying mechanism of how HxtR regulates the virulence of *S. suis*.

## 5. Conclusions

In this work, we constructed a transposon mutagenesis library of *S. suis* containing 1665 mutants and identified the Tn insertion sites. By using *G. mellonella* larvae infection assay, a total of 32 gene loci were identified from the Tn library that were involved in the virulence of *S. suis*. We further characterized a previously unreported VRG, *hxtR*, which showed that it had a significant influence on virulence in larvae and mice and affected bacterial growth and division. Overall, our study uncovers some new VRGs and provides valuable information for exploring *S. suis* pathogenesis.

## Figures and Tables

**Figure 1 microorganisms-10-00868-f001:**
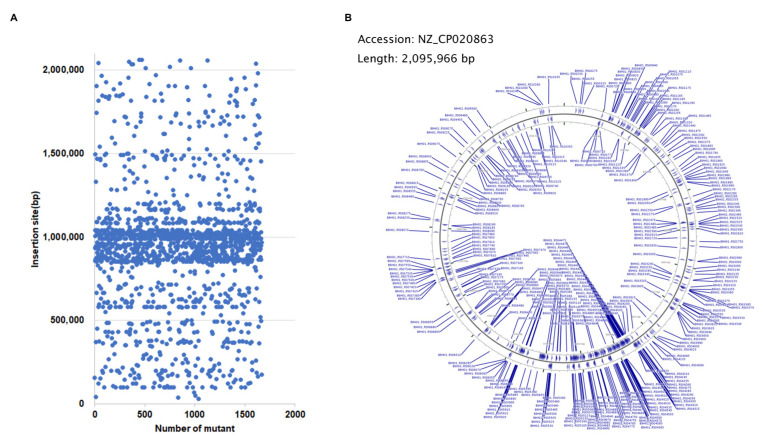
Location of transposon insertion sites in the genome of *S. suis* SC19. (**A**) Scatter plot of the positions of the Tn insertion sites. Blue spots represent the position (bp) of the 1665 mutants in the SC19 genome where the transposition inserted sites are located. (**B**) Mapping of the Tn insertion sites to the genome of *S. suis* SC19 strain. CGView Server was used to visualize the distribution of the inserted genes involved in the mutant library on the *S. suis* SC19 genome.

**Figure 2 microorganisms-10-00868-f002:**
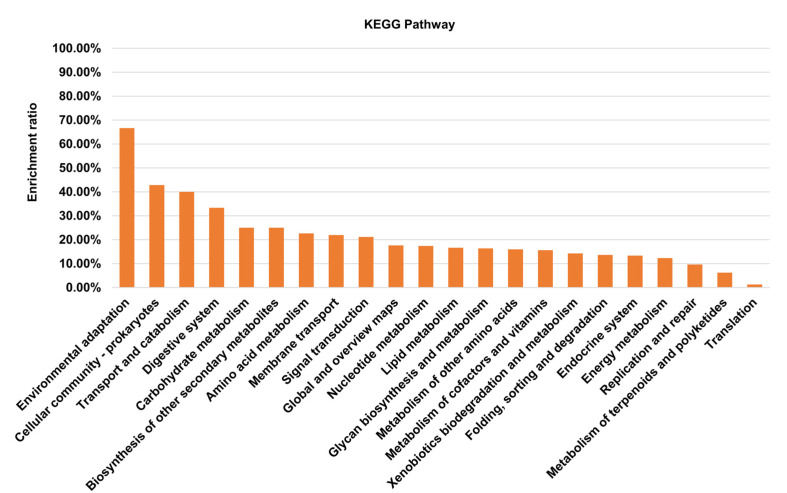
The KEGG pathway enrichment of the 417 Tn917-inserted genes. The 417 genes were assigned with a K number by using the BlastKOALA tool. The K number was then used to map each gene to its corresponding KEGG pathway using the Reconstruct tool of the KEGG mapper. The enrichment ratio indicates the number of genes identified from the Tn mutant library in each KEGG pathway to the number of total genes included in the *S. suis* SC19 genome in this pathway.

**Figure 3 microorganisms-10-00868-f003:**
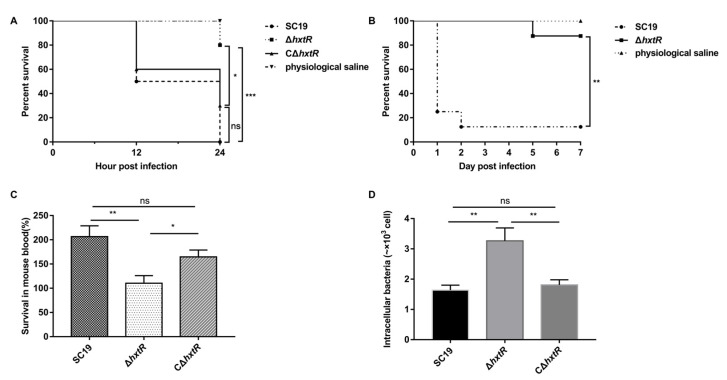
Virulence characterization of Δ*hxtR*, C*ΔhxtR,* and SC19 strains. (**A**) *G. mellonella* larvae infection assay; 20 μL of bacterial cells (approximately 3 × 10^6^ CFU) of Δ*hxtR*, CΔ*hxtR*, and SC19 strains at the mid-log phase were used to inject larva from the left posterior proleg. Each group contained 10 larvae. The survival was recorded at 12 and 24 hpi and statistically analyzed using the log-rank (Mantel–Cox) test. * indicates *p* < 0.05, and *** indicates *p* < 0.001. (**B**) Mouse infection assay; 500 μL of bacterial cells (8.5 × 10^8^ CFU) of Δ*hxtR* and SC19 strains at the mid-log phase were used to intraperitoneally inject mice. Each group contained 8 mice. The survival was recorded every 24 h post infection and statistically analyzed using the log-rank (Mantel–Cox) test. ** indicates *p* < 0.01. (**C**) Whole blood killing assay. Cells of Δ*hxtR*, CΔ*hxtR,* and SC19 strains at the mid-log phase were mixed with freshly prepared anticoagulated mouse blood, followed by incubation at 37 °C for 30 min. The mixture was then plated on TSA plates for viable bacteria enumeration. The survival was calculated as (CFU recovered/CFU in original inoculum) × 100% and statistically analyzed using the Student’s *t*-test. ns indicates *p* > 0.05, * indicates *p* < 0.05, and ** indicates *p* < 0.01. (**D**) Macrophage phagocytosis assay. Cells of Δ*hxtR*, CΔ*hxtR,* and SC19 strains were mixed with Raw264.7 macrophage cells with an MOI of 10:1, followed by incubation at 37 °C for 30 min. The cells were treated with ampicillin (50 μg/mL) to remove unphagocytized bacteria. The cells were then lysed, and the phagocytized bacteria were isolated and enumerated by plating on TSA plates. Statistical significances were determined by using the Student’s *t*-test. ns indicates *p* > 0.05, and ** indicates *p* < 0.01.

**Figure 4 microorganisms-10-00868-f004:**
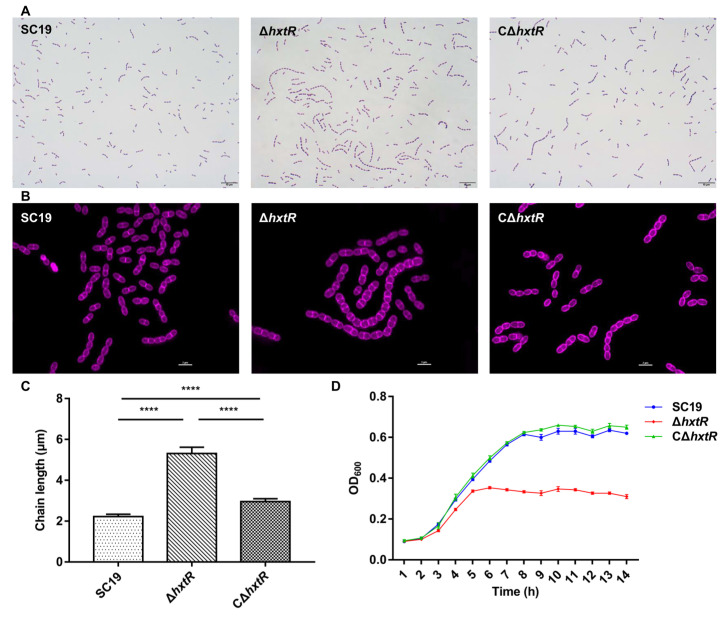
Cell morphology and growth characterizations of Δ*hxtR*, C*ΔhxtR,* and SC19 strains. (**A**) Morphological analysis by Gram staining. The cells of each indicated strain were grown to the mid-log phase, washed, subjected to regular Gram staining, and imaged under a light microscope. The scale bar is 10 μm. (**B**) Morphological analysis with SIM. The cells of each indicated strain were grown to the mid-log phase, washed, stained with fluorescent dye Alexa Fluor 647, and imaged using a two-color structure illumination microscope (SIM; Nikon Instruments, Tokyo, Japan) with excitation at 651 nm and emission at 672 nm. (**C**) Comparison of bacterial chain length. Over one hundred chains per sample were randomly selected in different fields from the Gram staining images to measure the length of the chain by using the software Image-Pro, and then the average chain length was calculated (in µm) and statistically analyzed using the Student’s *t*-test. **** indicates *p* < 0.0001. (**D**) Growth assay. The overnight cultures of the strains were diluted to an initial OD600 of 0.01 with TSB, and then, the diluted culture was pipetted into a microplate. The plate was incubated at 37 °C with continuous shaking for 14 h, and the optical density at 600 nm was monitored every 1 h.

**Table 1 microorganisms-10-00868-t001:** Bacterial strains, plasmids, and primers used in this study.

Strains, Plasmids, or Primers	Relevant Characteristics and/or Sequences	Source or Reference
**Strains**		
*S. suis* SC19	Virulent serotype 2 strain	[22]
Δ*hxtR*	A *hxtR* gene deletion mutant of *S. suis* SC19	This work
CΔ*hxtR*	A complementary strain of SC19 *Δ**hxtR*	This work
*E. coli* DH5α	F^-^*endA1 glnV44 thi-1 recA1 relA1 gyrA96 deoRnupG*Φ80d*lacZ*ΔM15 Δ(*lacZYA-argF*) U169, *hsdR17*(r _K_^-^m _K_^+^), λ-	Vazyme
**Plasmids**		
pTV408	For construction of *S. suis* transposon library (Kan^R^, Erm^R^, Amp^R^)	Dr. Tracy Wang
pSET4s	*E. coli-S. suis* shuttle vector (Spc^R^)	[23]
pSET4s-H	Derived from pSET4s used to knock out *hxtR* in SC19; Spc^R^	This work
pSET2	*E. coli-S. suis* shuttle vector (Spc^R^)	[24]
pSET2-CH	Derived from pSET2 for functional complementation of *hxtR* (Spc^R^)	This work
**Primers for identification of pTV408 transformant**		
JPM48	AATGCGGCCGCATGTCAGACATTTTAAA	This work
JPM49	TTTATCTGGAACATCTGTGG	This work
JPM50	AAGGGACCACCTATGATGTG	This work
JPM51	CAGAAGGCAATGTCATACCA	This work
**Primers for identification of *S. suis* SC19**		
16SrRNA-F	GTAGTCCACGCCGTAAACG	This work
16SrRNA-R	TAAACCACATGCTCCACCGC	This work
GDH-F	GCAGCGTATTCTGTCAAACG	This work
GDH-R	CCATGGACAGATAAAGATGG	This work
**Primers for Linker PCR**		
254	CGACTGGACCTGGA	This work
256	GATAAGCAGGGATCGGAACCTCCAGGTCCAGTCG	This work
Tn917-seq	AGAGAGATGTCACCGTCAAGT	This work
258	GATAAGCAGGGATCGGAACC	This work
**Primers for Δ*hxtR* and CΔ*hxtR***		
Δ*hxtR*-A	AATTCGAGCTCGGTACCCGGCATCTCCAGCATTTTCCTTC	This work
Δ*hxtR*-B	TAGGCTTAAAAATCATAAAATATCACCTAAAATCATGATTGTC	This work
Δ*hxtR*-C	TTAGGTGATATTTTATGATTTTTAAGCCTAGTTAATCACTAGT	This work
Δ*hxtR*-D	GCCTGCAGGTCGACTCTAGAGGGTTTCTGTAGAAGATTTTCCTA	This work
Δ*hxtR*-outF	CATGCTGACAGGATAGACATAGGA	This work
Δ*hxtR*-outR	AAGAGCAAGAATTTGGCATCG	This work
CΔ*hxtR*-F	AATTCGAGCTCGGTACCCGGTCGTAATCGGCTAATAAGTC	This work
CΔ*hxtR*-R	GCCTGCAGGTCGACTCTAGATCAATCTGGACTATAAATATCTACAA	This work
M13F	GTAAAACGACGGCCAGT	This work
M13R	CAGGAAACAGCTATGAC	This work
5150-F	ATGAAAAGGATTACAGAGATTTCTTG	This work
5150-R	TCATGTGAGAGGTTTTGACC	This work
*hxtR*-F	ATGATTTTAGGTGATATTTTAAAAGAATACCG	This work
*hxtR*-R	TCAATCTGGACTATAAATATCTACAACTT	This work
5160-F	ATGTGGCCGGAGGAAAAGA	This work
5160-R	TCAACTTTTTTGCTTTTCTTTTTCCTTGA	This work

Erm^R^—erythromycin resistant; Kan^R^—kanamycin resistant; Amp^R^—ampicillin resistant; Spc^R^—spectinomycin resistant.

**Table 2 microorganisms-10-00868-t002:** Transposants with reduced virulence in *Galleria mellonella* larvae.

Strain	Locus Tag	Product	Cumulative Mortality within 12, 18, and 24 hpi	Animal Model^ref.^
*S. suis* SC19		4/6; 6/6; 6/6		
**Transcription regulator**				
Tn1864	B9H01_RS05155	XRE family transcriptional regulator	1/6; 2/6; 2/6	
Tn491	B9H01_RS04445	DNA-binding response regulator	1/6; 1/6; 6/6	
**Transporter**				
Tn722	B9H01_RS07420	Voltage-gated chloride channel protein	2/6; 2/6; 3/6	
Tn1673	B9H01_RS04375	MULTISPECIES: ABC transporter ATP-binding protein	1/6; 3/6; 3/6	
Tn1	B9H01_RS00985	Sugar ABC transporter permease	0/6; 2/6; 4/6	
**Others**				
Tn1862	B9H01_RS01125	Formate C-acetyltransferase (PFL)	0/6; 0/6; 0/6	
Tn509	B9H01_RS05230	Peptidase	1/6; 2/6; 2/6	
Tn513	B9H01_RS05380	SAM-dependent methyltransferase	0/6; 1/6; 2/6	
Tn1712	B9H01_RS07385	Site-specific integrase	1/6; 2/6; 2/6	
Tn28	B9H01_RS04890	DUF1836 domain-containing protein	1/6; 1/6; 3/6	
Tn140	B9H01_RS05665	UDP-N-acetylglucosamine 1-carboxyvinyltransferase	2/6; 2/6; 3/6	
Tn271	B9H01_RS05640	Beta-glucuronidase	0/6; 1/6; 3/6	
Tn660	B9H01_RS05870	Hyaluronidase	0/6; 0/6; 0/6	Not tested [34]
Tn1803	B9H01_RS05855	Hyaluronidase	1/6; 3/6; 4/6	Not tested [34]
Tn4	B9H01_RS04330	Phosphomannomutase/phosphoglucomutase	0/6; 2/6; 4/6	
Tn98	B9H01_RS03990	Subtilisin-like serine protease (SspA)	1/6; 2/6; 4/6	Mouse and pig [35,36]
Tn220	B9H01_RS03140	Carbamate kinase (ArcC belongs to ADS)	0/6; 2/6; 2/6	Not tested [9,37]
Tn376	B9H01_RS10315	Glycosyl hydrolase	0/6; 1/6; 4/6	
Tn454	B9H01_RS05605	Beta-hexosamidase	0/6; 4/6; 4/6	
Tn496	B9H01_RS03375	DUF975 domain-containing protein	1/6; 3/6; 4/6	
Tn866	B9H01_RS04990	1,4-alpha-glucan branching protein (GlgB)	1/6; 3/6; 4/6	
Tn1823	B9H01_RS05160	A/G-specific adenine glycosylase	1/6; 3/6; 4/6	
Tn15	B9H01_RS04765	Glucan-binding protein	0/6; 5/6; 5/6	
Tn316	B9H01_RS05135	Integrase	1/6; 2/6; 5/6	
Tn656	B9H01_RS06130	Histidine triad protein	1/6; 3/6; 5/6	
Tn1761	B9H01_RS00820	Type II secretion pathway, pseudopilin (PulG)	1/6; 3/6; 5/6	
**Hypothetical protein**				
Tn1624	B9H01_RS03005	Hypothetical protein	0/6; 3/6; 4/6	
Tn122	B9H01_RS04405	Hypothetical protein	1/6; 2/6; 3/6	
Tn12	B9H01_RS00275	Hypothetical protein	1/6; 3/6; 4/6	
Tn29	B9H01_RS05145	Hypothetical protein	1/6; 3/6; 4/6	
Tn524	B9H01_RS04840	Hypothetical protein	1/6; 3/6; 4/6	
Tn983	B9H01_RS03005	Hypothetical protein	0/6; 4/6; 5/6	

Animal model indicates that the virulence of the gene was reported or the virulence was tested using animal infection models in *S. suis*.

## Data Availability

Not applicable.

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
