# Peer review of "Screening for Virulence-Related Genes via a Transposon Mutant Library of Streptococcus suis Serotype 2 Using a Galleria mellonella Larvae Infection Model"

_microorganisms, 2022, doi:10.3390/microorganisms10050868_

Round 1

Reviewer 1 Report

General comments

In this study, in order to find novel virulence-related genes (VRGs) of S. suis, the authors used a transposon mutagenesis technique and a Galleria mellonella larvae infection model. They found many possible VRGs and further characterized one of the VRGs. Their strategy based on traditional (transposon) and relatively novel (Galleria mellonella) techniques are useful to search for VRGs, so may be worth publishing. However, I have many concerns on the manuscript as described below.

Major comments

  1. L93: Where do primers JPM48, 49, 50 and 51 anneal? Are they the primers to detect parts of the transposon?
  2. L133: How did the authors extract RNA used for RT-PCR?
  3. L113 and L139-140: In this study, the authors adjust concentration of bacterial suspensions for infection experiments by using MacFarland Standards. Infection dose is one of the most important factors that affect outcome of infection experiments, so it is necessary to prepare the same concentration of inocula for accurate comparison of virulence. However, generally, it is difficult to adjust bacterial concentration accurately by visual comparison using MacFarland. Why didn’t the authors adjust the concentration by measuring OD600?
  4. L157: Did the authors wash cells after the ampicillin treatment?
  5. The authors should explain statistical analysis methods in detail (including software used) in the Materials and Methods section.
  6. In Figure 2, the authors show KEGG pathway analysis results. However, I could not find methods for the analysis in the Materials and Methods section.
  7. L197-201 (Figure 2 legend): Explain what the blue and orange colors in the graph mean. Also, I could not understand exact meaning of “the percentage of the pathway involved in the inserted gene to the background pathway” in L200-201. What is the background pathway? Rework the sentence.
  8. L204: In this study, the authors successfully disrupted 417 different genes, but only 391 mutants were tested by the G. mellonella model. How did the authors select 391 mutants from the 417 mutants?
  9. L204-205: To what did the authors compare the results of mutants when the statistical analyses were done? To the results of SC-19-infected larvae?
  10. According to Table 2, strain Tn491 killed all larvae within 24 h. Strains Tn15, Tn316, Tn656, Tn1761 and Tn983 killed 5 of 6 larvae within 24 h. I wonder if they are really attenuated. Maybe, more larvae survived at 12 and 18 hpi, and thus significant differences were detected by the log-rank tests. But to convince readers, the authors should also show the mortality of all mutant strains at 12 hpi and 18hpi in Table 2. In addition, for reference, it is necessary to show the results of SC-19-infected larvae in Table 2 too.
  11. L216-217: The authors claimed that the knockout of hxtR did not affect the hxtR upstream and downstream genes transcriptions. However, Fig. S1B suggests that the knockout of hxtR caused increased transcription of B9H01_RS05150. So, the authors need to discuss the influence of increased expression of B9H01_RS05150 on the outcome of experimental infections.
  12. Figure 3A: The "ns" in this figure should be "*" because it is described that the mutant showed significantly attenuated virulence compared with CΔhxtR (p<0.05) in L230-231,.
  13. In L239-240, it is described that the ΔhxtR displayed declined survival in Raw264.7 cells compared with SC-19. However, Fig. 3D shows that ΔhxtR survived more in Raw264.7 cells than SC-19. That is, the figure and explanation are inconsistent! I guess Figure 3D is not the figure to show the survival ability of the strains in Raw264.7 cells but the one to show phagocytosis resistant ability of the strains. So, the authors should rework the legend.
  14. Figure 4B is too dark to show the morphological differences.
  15. Please discuss why ΔhxtR was attenuated in larvae and mice (i.e., what had happened in the insects and animals) on the basis of the results of in vitro studies.

Minor corrections or comments

  1. L43: I don’t think MRP is a virulence factor but just a virulence marker or a virulence-associated factor because MRP defective mutants did not affect pigs (DOI: 10.1128/iai.64.10.4409-4412.1996 ).
  2. L46: VRGs → virulence-related genes (VRGs)
  3. L52: Salmonella gallinarumSalmonella Gallinarum
  4. I found SC19 and SC-19 in this manuscript. Are they the same strain? If yes, which is the correct name?
  5. L99: Describe how many units of AluI did authors use for the reaction.
  6. L133: RT-PCR → reverse transcription (RT)-PCR
  7. L173: Cite a paper after "as described previously".
  8. L174: Morphology observation → Growth measurements
  9. L182: Mutants production → Mutants construction
  10. L186: Write the accession number of SC-19 genome after "the genome".
  11. L188: What is the transgenic region??? Did the authors want to say "the intergenic region" ?
  12. L250: (Fig. 4C) → (Fig. 4ABC)
  13. Figure S1: Add lane numbers in the figure.

Reviewer 2 Report

Authors prepared transposon mutant library of Streptococcus suis in order to screen virulence related genes of S. suis. Authors identified mutants in 417 genes, 32 of them were attenuated in Galleria mellonella infection model. 30 novel virulence related genes were discovered. Deletion mutant in one of them, the hxtR gene, was studied in details for growth, morphology, whole blood bactericidal assay, phagocytosis and mouse infection model. Attenuation of hxtR gene mutant was confirmed in all assays. Authors proved transcriptional regulator hxtR is involved in virulence of S. suis.

In my opinion, the work was performed well, methods used were appropriate for such kind of experiment. Data presentation is clear and conclusion are in agreement with data obtained. I have no major comments and I recommend to accept manuscript after minor revision.

Minor comments

Please report agreement for mice experiment.

Add word “groups” in the sentence on line 138. It probably should reads “grouped into three groups (n=8)” Does it mean, that each group consisted of eight animals? Please clarify.

Consider using word “intergenic” on line 188 instead of “transgenic”.

Figure 3 caption start with sentence “This is a figure.” What does it mean?

“Alexa Fluor” instead of “Alexa Flour” on lines 249 and 257.

Round 2

Reviewer 1 Report

The manuscript has been improved. I have only several minor suggestions.

L93: Please spell DfSB out.

L100: All the positive colonies → All the erythromycin resistance cassette-positive and kanamycin resistance cassette-negative colonies

L155: 5160F/R → 5160F/5160R

L296: (CFU) → (approximately 3 × 106 CFU)

L390-391 and 394: Please write which animal models the previous studies used to show attenuated virulence of S. aureus and S. pneumoniae.

Author Response

Dear reviewer 1

We thank the reviewer for the very helpful suggestions, and we have revised them.

We are looking forward to hearing from you.

Sincerely yours,

Dr. Qi Huang

Associate Professor,

College of Veterinary Medicine,

Huazhong Agricultural University,

Wuhan 430070, China

E-mail: qhuang@mail.hzau.edu.cn